# Long Short-Term Transformer for Online Action Detection

**Mingze Xu**      **Yuanjun Xiong**      **Hao Chen**      **Xinyu Li**
**Wei Xia**      **Zhuowen Tu**      **Stefano Soatto**
Amazon/AWS AI
{xumingze,yuanjx,hxen,xxnl,wxia,ztu,soattos}@amazon.com

## Abstract

We present Long Short-term TRansformer (LSTR), a temporal modeling algorithm for online action detection, which employs a long- and short-term memory mechanism to model prolonged sequence data. It consists of an LSTR encoder that dynamically leverages coarse-scale historical information from an extended temporal window (*e.g.*, 2048 frames spanning of up to 8 minutes), together with an LSTR decoder that focuses on a short time window (*e.g.*, 32 frames spanning 8 seconds) to model the fine-scale characteristics of the data. Compared to prior work, LSTR provides an effective and efficient method to model long videos with fewer heuristics, which is validated by extensive empirical analysis. LSTR achieves state-of-the-art performance on three standard online action detection benchmarks, THUMOS'14, TVSeries, and HACS Segment. Code has been made available at: `https://xumingze0308.github.io/projects/lstr`.

## 1 Introduction

Given an incoming stream of video frames, online action detection [14] is concerned with the task of classifying what is happening at each frame without seeing the future. Unlike offline methods that assume the entire video is available, online methods process the data causally, up to the current time. In this paper, we present an online temporal modeling algorithm capable of capturing temporal relations on prolonged sequences up to 8 minutes long, while retaining fine granularity of the event in the representation. This is achieved by modeling activities at different temporal scales, so as to capture a variety of events ranging from bursts to slow trends.

Specifically, we propose a method, named *Long Short-term TRansformer (LSTR)*, to jointly model long- and short-term temporal dependencies. LSTR has two main advantages over prior work. 1) It stores the history directly thus avoiding the pitfalls of recurrent models [18, 50, 28, 10]. Back-propagation through time, BPTT, is not needed as the model can directly attend to any useful frames from memory. 2) It separates long- and short-term memories, which allows modeling short-term context while extracting useful correlations from the long-term history. This allows us to compress the long-term history without losing important fine-scale information.

As shown in Fig. 1, we explicitly divide the entire history into the long- and short-term memories and build our model with an encoder-decoder architecture. Specifically, the *LSTR encoder* compresses and abstracts the long-term memory into a latent representation of fixed length, and the *LSTR decoder* uses a short window of transient frames to perform self-attention and cross-attention operations on the extracted token embeddings from the LSTR encoder. In the LSTR encoder, an extended temporal support becomes beneficial in dealing with untrimmed, streaming videos by devising two-stage memory compression, which is shown to be computationally efficient in both training and inference. Our overall long short-term Transformer architecture gives rise to an effective and efficient representation for modeling prolonged sequence data.

35th Conference on Neural Information Processing Systems (NeurIPS 2021).

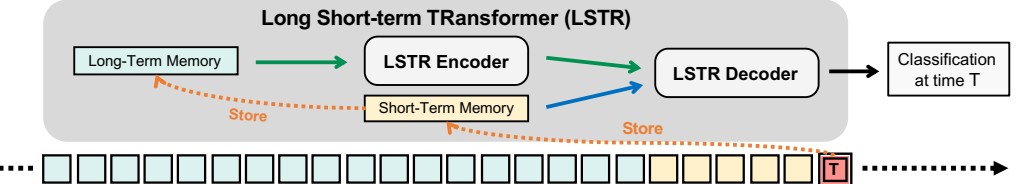

Figure 1: **Overview of Long Short-term TRansformer (LSTR)**. Given a live streaming video, LSTR sequentially identifies the actions happening in each incoming frame by using an encoder-decoder architecture, without future context. The dashed brown arrows indicate the data flow of the long- and short-term memories following the first-in-first-out (FIFO) logic. (Best viewed in color.)

We validate LSTR on standard benchmark datasets (THUMOS'14 [30], TVSeries [14], and HACS Segment [76]). These have distinct characteristics such as video length spanning from a few seconds to tens of minutes. Experimental results establish LSTR as the state-of-the-art for online action detection. Ablation studies further showcase LSTR's abilities in modeling long video sequences.

## 2 Related Work

**Online Action Detection.** Temporal action localization aims to detect the onset and termination of action instances after observing the entire video [52, 71, 23, 53, 78, 5, 37, 39, 38]. Embodied perception, however, requires causal processing [73], where we can only process the data up to the present. Online action detection focuses on this setting [14]. RED [22] uses a reinforcement loss to encourage recognizing actions as early as possible. TRN [72] models greater temporal context by simultaneously performing online action detection and anticipation. IDN [19] learns discriminative features and accumulates only relevant information for the present. LAP-Net [46] proposes an adaptive sampling strategy to obtain optimal features. PKD [77] transfers knowledge from offline to online models using curriculum learning. As with early action detection [27, 40], Shou *et al.* [54] focus on online detection of action start (ODAS). StartNet [24] decomposes ODAS into two stages and learns with policy gradient. WOAD [25] uses weakly-supervised learning with video-level labels.

**Temporal/Sequence Modeling.** Causal time series analysis has traditionally assumed the existence of a latent "state" variable that captures all information in past data, and is updated using only the current datum [49, 28, 10]. While the Separation Principle ensures that such a state exists for linear-Gaussian time series, in general it is not possible to summarize all past history of complex data in a finite-dimensional sufficient statistic. Therefore, we directly model the history, in accordance with other work on video understanding [69, 45, 68]. Earlier work on action recognition usually relies on heuristic sub-sampling (typically 3 to 7 video frames) for more feasible training [74, 62, 21, 41, 57]. 3D ConvNets [59, 8, 60] are used to perform spatio-temporal feature modeling on more frames, but they fail to capture temporal correlations beyond their receptive field. Recently, Wu *et al.* [69] propose long-term feature banks to capture objects and scene features, but discarding their temporal order which is clearly informative. Most of work above does not explicitly separate the long- and short-term context modeling, but instead integrates all observed features with simple mechanisms such as pooling or concatenation. We are motivated by work in Cognitive Science [44, 12, 9, 35] that has shed light on the design principles for modeling long-term dependencies with attention mechanism [66, 13, 48, 75].

**Transformers for Action Understanding.** Transformers have achieved breakthrough success in NLP [47, 15] and are adopted in computer vision for image recognition [17, 58] and object detection [7]. Recent papers exploit Transformers for temporal modeling tasks in videos, such as action recognition [43, 51, 36, 4, 3] and temporal action localization [42, 56], and achieve promising results. However, computational and memory demands result in most work being limited to short video clips, with few exceptions [13, 6] that focuses on designing Transformers to model long-range context. The mechanism for aggregating long- and short-term information is relatively unexplored [32].

## 3 Long Short-Term Transformer

Given a live streaming video, our goal is to identify the actions performed in each video frame using only past and current observations. Future information is *not* accessible during inference. Formally, a streaming video at time $t$ is represented by a batch of $\tau$ past frames $\mathbf{I}^t = \{I_{t-\tau}, \cdots, I_t\}$,

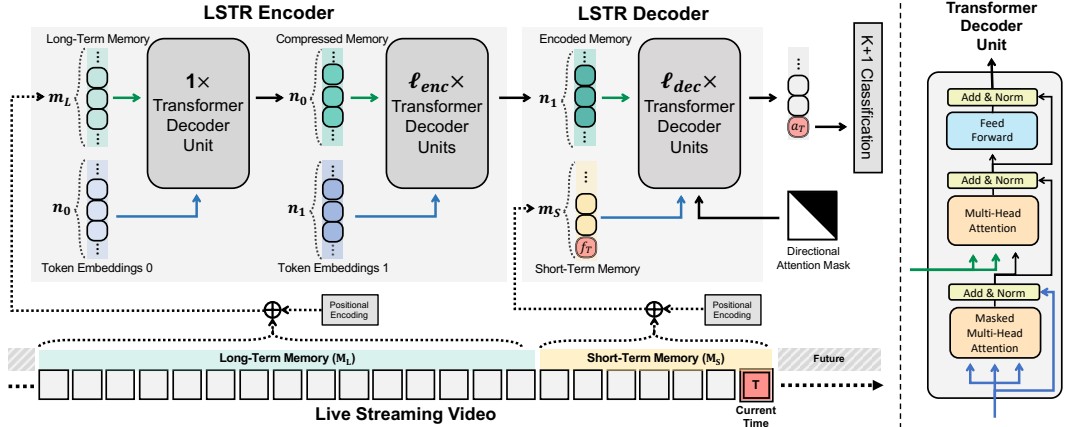

Figure 2: **Visualization of Long Short-Term Transformer (LSTR)**, which is formulated in an encoder-decoder manner. Specifically, the LSTR encoder compresses the long-term memory of size $m_L$ to $n_1$ encoded latent features, and the LSTR decoder references related context information from the encoded memory with the short-term memory of size $m_S$ for action recognition of the present. The LSTR encoder and decoder are built with Transformer decoder units [61], which take the input tokens (dark green arrows) and output tokens (dark blue arrows) as inputs. During inference, LSTR processes every incoming frame in an online manner, absent future context. (Best viewed in color.)

which reads "$I$ up to time $t$." The online action detection system receives $\mathbf{I}^t$ as input, and classifies the action category $\hat{y}_t$ belonging to one of $(K + 1)$ classes, $\hat{y}_t \in \{0, 1, \cdots, K\}$, ideally using the posterior probability $P(\hat{y}_t = k | \mathbf{I}^t)$, where $k = 0$ denotes the probability that no event is occurring at frame $t$. We design our method by assuming that there is a pretrained feature extractor [62] that processes each video frame $I_t$ into a feature vector $\mathbf{f}_t \in \mathbb{R}^C$ of $C$ dimensions[1]. These vectors form a $(\tau \times C)$-dimensional temporal sequence that serves as the input of our method.

## 3.1 Overview

Our method is based on the intuition that frames observed recently provide precise information about the ongoing action instance, while frames over an extended period offer contextual references for actions that are potentially happening right now. We propose Long Short-term TRansformer (LSTR) in an explicit encoder-decoder manner, as shown in Fig. 2. In particular, the feature vectors of $m_L$ frames in the distant past are stored in a *long-term memory*, and a *short-term memory* stores the features of $m_S$ recent frames. The LSTR encoder compresses and abstracts features in long-term memory to an encoded latent representation of $n_1$ vectors. The LSTR decoder queries the encoded long-term memory with the short-term memory for decoding, leading to the action prediction $\hat{y}_t$. This design follows the line of thought in combining long- and short-term information for action understanding [16, 62, 69], but addresses several key challenges to efficiently achieve this goal, exploiting the flexibility of Transformers [61].

## 3.2 Long- and Short-Term Memories

We store the streaming input of feature vectors into two consecutive memories. The first is the short-term memory which stores only a small number of frames that are recently observed. We implement it with a first-in-first-out (FIFO) queue of $m_S$ slots. At time $T$, it stores the feature vectors as $\mathbf{M}_S = \{\mathbf{f}_T, \cdots, \mathbf{f}_{T-m_S+1}\}$. When a frame becomes "older" than $m_S$ time steps, it graduates from $\mathbf{M}_S$ and enters into the long-term memory, which is implemented with another FIFO queue of $m_L$ slots. The long term memory stores $\mathbf{M}_L = \{\mathbf{f}_{T-m_S}, \cdots, \mathbf{f}_{t-m_S-m_L+1}\}$. The long-term memory serves as the input memory to the LSTR encoder and the short-term memory serves as the queries for the LSTR decoder. In practice, the long-term memory stores a much longer time span than the short-term memory ($m_S \ll m_L$). A typical choice is $m_L = 2048$, which represents 512 seconds worth of video contents with 4 frames per second (FPS) sampling rate, and $m_S = 32$ representing

---

[1]In practice, some feature extractors [59] take consecutive frames to produce one feature vector. Nonetheless, it is still temporally "centered" on a single frame. Thus we use the single frame notation here for simplicity.

8 seconds. We add a sinusoidal positional encoding $\mathbf{s}$ [61] to each frame feature in the memories *relative* to current time $T$ (*i.e.*, the frame at $T - \tau$ receives a positional embedding of $\mathbf{s}_\tau$).

## 3.3 LSTR Encoder

The LSTR encoder aims at encoding the long-term memory of $m_L$ feature vectors into a latent representation that LSTR can use for decoding useful temporal context. This task requires a large capacity in capturing the relations and temporal context across a span of hundreds or even thousands of frames. Prior work on modeling long-term context for action understanding relies on heuristic temporal sub-sampling [69, 62] or recurrent networks [16] to make training feasible, at the cost of losing specific information of each time step. Attention-based architectures, such as Transformer [61], have recently been shown promising for similar tasks that require long-range temporal modeling [51]. A straightforward choice for LSTR encoder would be to use a Transformer encoder based on self-attention. However, its time complexity, $O(m_L^2 C)$, grows quadratically with the memory sequence length $m_L$. This limits our ability to model long-term memory with sufficient length to cover long videos. Though recent work [64] has explored self-attention with linear complexity, repeatedly referencing information from the long-term memory with multi-layer Transformers is still computationally heavy. In LSTR, we propose to use the two-stage memory compression mechanism based on Transformer decoder units [61] to achieve more effective memory encoding.

**The Transformer decoder unit** [61] takes two sets of inputs. The first set includes a fixed number of $n$ learnable output tokens $\boldsymbol{\lambda} \in \mathbb{R}^{n \times C}$, where $C$ is the embedding dimension. The second set includes another $m$ input tokens $\boldsymbol{\theta} \in \mathbb{R}^{m \times C}$, where $m$ can be a rather large number. It first applies one layer of multi-head self-attention on $\boldsymbol{\lambda}$. The outputs $\boldsymbol{\lambda}'$ are then used as queries in an "QKV cross-attention" operation and the input embeddings $\boldsymbol{\theta}$ serve as key and value. The two steps can be written as

$$\boldsymbol{\lambda}' = \text{SelfAttn}(\boldsymbol{\lambda}) = \text{Softmax}(\frac{\boldsymbol{\lambda} \cdot \boldsymbol{\lambda}^T}{\sqrt{C}})\boldsymbol{\lambda} \;\; and \;\; \text{CrossAttn}(\sigma(\boldsymbol{\lambda}'), \boldsymbol{\theta}) = \text{Softmax}(\frac{\sigma(\boldsymbol{\lambda}') \cdot \boldsymbol{\theta}^T}{\sqrt{C}})\boldsymbol{\theta},$$

where $\sigma : \mathbb{R}^{n \times C} \to \mathbb{R}^{n \times C}$ denotes the intermediate layers between the two attention operations. One appealing property of this design is that it transforms the $m \times C$ dimensional input tokens into output tokens of $n \times C$ dimensions in $O(n^2 C + nmC)$ time complexity. When $n \ll m$, the time complexity becomes linear to $m$, making it an ideal candidate for compressing long-term memory. This property is also utilized in [32] to efficiently process large volume inputs, such as image pixels.

**Two-Stage Memory Compression.** Stacking multiple Transformer decoder units on the long-term memory, as in [61], can form a memory encoder with linear complexity with respect to the memory size $m_L$. However, running the encoder at each time step can still be time consuming. We can further reduce the time complexity with a two-stage memory compression design. The first stage has one Transformer decoder unit with $n_0$ output tokens. Its input tokens are the entire long-term memory of size $m_L$. The outputs of the first stage are used as the input tokens to the second stage, which has $\ell_{enc}$ stacked Transformer decoder units and $n_1$ output tokens. Then, the long-term memory of size $m_L \times C$ is compressed into a latent representation of size $n_1 \times C$, which can then be efficiently queried in the LSTR decoder later. This two-stage memory compression design is illustrated in Fig. 2.

Compared to an $(1 + \ell_{enc})$-layer Transformer encoder with $O(m_L^2 (1 + \ell_{enc})C)$ time complexity or stacked Transformer decoder units with $n$ output-tokens having $O((n^2 + nm_L)(1 + \ell_{enc})C)$ time complexity, the proposed LSTR encoder has complexity of $O(n_0^2 C + n_0 m_L C + (n_1^2 + n_1 n_0)\ell_{enc}C)$. Because both $n_0$ and $n_1$ are much smaller than $m_L$, and $\ell_{enc}$ is usually larger than 1, using two-stage memory compression could be more efficient. In Sec. 3.6, we will show that, during online inference, it further enables us to reduce the runtime of the Transformer decoder unit of the first stage. In Sec. 4.5, we empirically found this design also leads to better performance for online action detection.

## 3.4 LSTR Decoder

The short-term memory contains informative features for classifying actions on the latest time step. The LSTR decoder uses the short-term memory as queries to retrieve useful information from the encoded long-term memory produced by the LSTR encoder. The LSTR decoder is formed by stacking $\ell_{dec}$ layers of Transform decoder units. It takes the outputs of the LSTR encoder as input tokens and the $m_S$ feature vectors in the short-term memory as output tokens. It outputs $m_S$ probability vectors $\{\mathbf{p}_T, \cdots, \mathbf{p}_{T-m_S+1}\} \in [0, 1]^{K+1}$, each $\mathbf{p}_t$ representing the predicted probability distribution of $K$ action categories and one "background" class at time $t$. During inference, we only take the probability vector $\mathbf{p}_T$ from the output token corresponding to the current time $T$ for the classification result.

However, having the additional outputs on the older frames allows the model to leverage more supervision signals during training. The details will be described below.

## 3.5   Training LSTR

LSTR can be trained without temporal unrolling and Backpropagation Through Time (BPTT) as in LSTM [28], which is a common property of Transformers [61]. We construct each training sample by randomly sampling an ending time $T$ and filling the long- and short-term memories by tracing back in time for $m_S + m_L$ frames. We use the empirical cross entropy loss between the predicted probability distribution $\mathbf{p}_T$ at time $T$ and the ground truth action label $y_T \in \{0, 1, \cdots, K\}$ as

$$L(y_T, \mathbf{p}_T; T) = -\sum_{k=0}^{K} \delta(k - y_T) \log p_T^k, \tag{1}$$

where $p_T^k$ is the $k$-th element of the probability vector $\mathbf{p}_T$, predicted on the latest frame at $T$. Additionally, we add a *directional attention mask* [61] to the short-term memory so that any frame in the short-term memory can only depend on its previous frames. In this way, we can make predictions on all frames in the short-term memory as if they are the latest ones. Thus we can provide supervision on every frame in the short-term memory, and the complete loss function $\mathcal{L}$ is then

$$\mathcal{L}_T = \sum_{t=T-ms+1}^{T} L(y_t, \mathbf{p}_t; T), \tag{2}$$

where $\mathbf{p}_t$ denotes the prediction from the output token corresponding to time $t$.

## 3.6   Online Inference with LSTR

During online inference, the video frame features are streamed to the model as time passes. Running LSTR's long-term memory encoder from scratch for each frame results in a time complexity of $O(n_0^2 C + n_0 m_L C)$ and $O((n_1^2 + n_1 n_0)\ell_{enc} C)$ for the first and second memory compression stages, respectively. However, at each time step, there is only one new video frame to be updated. We show it is possible to achieve even more efficient online inference by storing the intermediate results for the Transformer decoder unit of the first stage. First, the queries of the first Transformer decoder unit are fixed. So their self-attention outputs can be pre-computed and used throughout the inference. Second, the cross-attention operation in the first stage can be written as

$$\text{CrossAttn}(\mathbf{q}_i, \{\mathbf{f}_{T-\tau} + \mathbf{s}_\tau\}) = \sum_{\tau=m_S}^{m_S+m_L-1} \frac{\exp((\mathbf{f}_{T-\tau} + \mathbf{s}_\tau) \cdot \mathbf{q}_i / \sqrt{C})}{\sum_{\tau=m_S}^{m_S+m_L-1} \exp((\mathbf{f}_{T-\tau} + \mathbf{s}_\tau) \cdot \mathbf{q}_i / \sqrt{C})} \cdot (\mathbf{f}_{T-\tau} + \mathbf{s}_\tau), \tag{3}$$

where the index $\tau = T - t$ is the relative position of a frame $t$ in the long-term memory to the latest time $T$. This calculation depends on the un-normalized attention weight matrix $\mathbf{A} \in \mathbb{R}^{m_L \times n_0}$, with elements $a_{\tau i} = (\mathbf{f}_{T-\tau} + \mathbf{s}_\tau) \cdot \mathbf{q}_i$. $\mathbf{A}$ can be decomposed into the sum of two matrices $\mathbf{A}^f$ and $\mathbf{A}^s$. We have their elements as $a_{\tau i}^f = \mathbf{f}_{T-\tau} \cdot \mathbf{q}_i$ and $a_{\tau i}^s = \mathbf{s}_\tau \cdot \mathbf{q}_i$. The queries after the first self-attention operation, $\mathbf{Q} = [\mathbf{q}_1, \ldots, \mathbf{q}_{n_0}]$, and the position embedding $\mathbf{s}_\tau$ are fixed during inference. Thus the matrix $\mathbf{A}^s$ can be pre-computed and used for every incoming frame. We additionally maintain a FIFO queue of vectors $\mathbf{a}_t = \mathbf{Q}^\top \mathbf{f}_t$ of size $m_L$. $\mathbf{A}^f$ at any time step $T$ can be obtained by stacking all vectors currently in this queue. Updating this queue at each time step requires $O(n_0 C)$ time complexity for the matrix-vector product. Now we can obtain the matrix $\mathbf{A}$ with only $n_0 \times m_L$ additions by adding $\mathbf{A}^s$ and $\mathbf{A}^f$ together, instead of $n_0 \times m_L \times C$ multiplications and additions using Eq. (3). This means the amortized time complexity for computing the attention weights can be reduced to $O(n_0(m_L + C))$. Although the time complexity of the cross-attention operation is still $O(n_0 m_L C)$ due to the inevitable operation of weighted sum, since $C$ is usually larger than 1024 [26], this is still a considerable reduction of runtime. LSTR's walltime efficiency is discussed in Sec. 4.6.

## 4   Experiments

### 4.1   Datasets

We evaluate our model on three publicly-available datasets: **THUMOS'14** [30], **TVSeries** [14] and **HACS Segment** [76]. THUMOS'14 includes over 20 hours of sports video annotated with 20 actions. We follow prior work [72, 19] and train on the validation set (200 untrimmed videos) and evaluate on the test set (213 untrimmed videos). TVSeries contains 27 episodes of 6 popular TV series, totaling 16 hours of video. The dataset is annotated with 30 realistic, everyday actions (*e.g.*, open door). HACS Segment is a large-scale dataset of web videos. It contains 35,300 untrimmed videos over 200 human action classes for training and 5,530 untrimmed videos for validation.

## 4.2 Settings

**Feature Encoding.** We follow the experimental settings of state-of-the-art methods [72, 19]. We extract video frames at 24 FPS and set the video chunk size to 6. Decisions are made at the chunk level, and thus accuracy is evaluated at every 0.25 second. For feature encoding, we adopt the TSN [62] models implemented in an open-source toolbox [11]. Specifically, the features are extracted by one visual model with the ResNet-50 [26] architecture from the central frame of each chunk and one motion model with the BN-Inception [31] architecture from the stacked optical flow fields between 6 consecutive frames [62]. The visual and motion features are concatenated along the channel dimension as the final feature **f**. We experiment with feature extractors pretrained on two datasets, ActivityNet and Kinetics.

**Implementation Details.** We implemented our proposed model in PyTorch [1], and performed all experiments on a system with 8 Nvidia V100 graphics cards. For all Transformer units, we set their number of heads as 16 and hidden units as 1024 dimensions. To learn model weights, we used the Adam [34] optimizer with weight decay $5 \times 10^{-5}$. The learning rate was linearly increased from zero to $5 \times 10^{-5}$ in the first $2/5$ of training iterations and then reduced to zero following a cosine function. Our models were optimized with batch size of 16, and the training was terminated after 25 epochs.

**Evaluation Protocols.** We follow prior work and use per-frame **mean average precision (mAP)** to evaluate the performance of online action detection. We also use per-frame **calibrated average precision (cAP)** [14] that was proposed for TVSeries to correct the imbalance between positive and negative samples, $cAP = \sum_k cPrec(k) * I(k)/P$, where $cPrec = TP/(TP + FP/w)$, $I(k)$ is 1 if frame $k$ is a true positive, $P$ is the number of true positives, and $w$ is the negative and positive ratio.

## 4.3 Comparison with the State-of-the-art Methods

Table 1: **Online action detection and anticipation results** on THUMOS'14 and TVSeries in terms of mAP and cAP, respectively. For online action detection, LSTR outperforms the state-of-the-art methods on THUMOS'14 by 3.7% and 2.4% in mAP and on TVSeries by 2.8% and 2.7% in cAP, using ActivityNet and Kinetics pretrained features, respectively. LSTR also achieves promising results for action anticipation. *Results are reproduced by us using their papers' default settings.

(a) **Results of online action detection using ActivityNet features**

|  | THUMOS'14 | TVSeries |
| --- | --- | --- |
|  | mAP (%) | mcAP (%) |
| CDC [53] | 44.4 | - |
| RED [22] | 45.3 | 79.2 |
| TRN [72] | 47.2 | 83.7 |
| FATS [33] | 51.6 | 81.7 |
| IDN [19] | 50.0 | 84.7 |
| LAP [46] | 53.3 | 85.3 |
| TFN [20] | 55.7 | 85.0 |
| LFB* [69] | 61.6 | 84.8 |
| **LSTR** (ours) | **65.3** | **88.1** |

(b) **Results of online action detection using Kinetics features**

|  | THUMOS'14 | TVSeries |
| --- | --- | --- |
|  | mAP (%) | mcAP (%) |
| FATS [33] | 59.0 | 84.6 |
| IDN [19] | 60.3 | 86.1 |
| TRN [72] | 62.1 | 86.2 |
| PKD [77] | 64.5 | 86.4 |
| WOAD [25] | 67.1 | - |
| LFB* [69] | 64.8 | 85.8 |
| **LSTR** (ours) | **69.5** | **89.1** |

(c) **Results of action anticipation using ActivityNet features**

|  | THUMOS'14 | TVSeries |
| --- | --- | --- |
|  | mAP (%) | mcAP (%) |
| EFC [22] | 34.4 | 72.5 |
| ED [22] | 36.6 | 74.5 |
| RED [22] | 37.5 | 75.1 |
| TRN [72] | 38.9 | 75.7 |
| TTM [65] | 40.9 | 77.9 |
| LAP [46] | 42.6 | 78.7 |
| **LSTR** (ours) | **50.1** | **80.8** |

Table 2: **Online action detection results when only portions of videos are considered** in cAP (%) on TVSeries (*e.g.*, 80%-90% means only frames of this range of action instances were evaluated). LSTR outperforms existing methods at every time stage, especially on boundary locations.

|  | Features | Portion of Video | | | | | | | | | |
| --- | --- | --- | --- | --- | --- | --- | --- | --- | --- | --- | --- |
|  |  | 0-10% | 10-20% | 20-30% | 30-40% | 40-50% | 50-60% | 60-70% | 70-80% | 80-90% | 90-100% |
| TRN [72] |  | 78.8 | 79.6 | 80.4 | 81.0 | 81.6 | 81.9 | 82.3 | 82.7 | 82.9 | 83.3 |
| IDN [19] | ActivityNet | 80.6 | 81.1 | 81.9 | 82.3 | 82.6 | 82.8 | 82.6 | 82.9 | 83.0 | 83.9 |
| TFN [20] |  | 83.1 | 84.4 | 85.4 | 85.8 | 87.1 | 88.4 | 87.6 | 87.0 | 86.7 | 85.6 |
| **LSTR** (ours) |  | **83.6** | **85.0** | **86.3** | **87.0** | **87.8** | **88.5** | **88.6** | **88.9** | **89.0** | **88.9** |
| IDN [19] |  | 81.7 | 81.9 | 83.1 | 82.9 | 83.2 | 83.2 | 83.2 | 83.0 | 83.3 | 86.6 |
| PKD [77] | Kinetics | 82.1 | 83.5 | 86.1 | 87.2 | 88.3 | 88.4 | 89.0 | 88.7 | 88.9 | 87.7 |
| **LSTR** (ours) |  | **84.4** | **85.6** | **87.2** | **87.8** | **88.8** | **89.4** | **89.6** | **89.9** | **90.0** | **90.1** |

We compare LSTR against other state-of-the-art methods [72, 19, 25] on THUMOS'14, TVSeries, and HACS Segment. Specifically, on THUMOS'14 and TVSeries, we implement LSTR with the long- and short-term memories of 512 and 8 seconds, respectively. On HACS Segment, we reduce

the long-term memory to 256 seconds, considering that its videos are strictly shorter than 4 minutes. For LSTR, we implement the two-stage memory compression using Transformer decoder units. We set the token numbers to $n_0 = 16$ and $n_1 = 32$ and the Transformer layers to $\ell_{enc} = 2$ and $\ell_{dec} = 2$.

### 4.3.1 Online Action Detection

**THUMOS'14.** We compare LSTR with recent work on THUMOS'14, including methods that use 3D ConvNets [53] and RNNs [70, 19, 25], reinforcement learning [22], and curriculum learning [77]. Table 1a and 1b shows that LSTR significantly outperforms the the state-of-the-art methods [69, 25] by 3.7% and 2.4% in terms of mAP using ActivityNet and Kinetics pretrained features, respectively.

**TVSeries.** Table 1a and 1b show the online action detection results that LSTR outperforms the state-of-the-art methods [20, 77] by 2.8% and 2.7% in terms of cAP using ActivityNet and Kinetics pretrained features, respectively. Following prior work [14], we also investigate LSTR's performance at different action stages by evaluating each decile (ten-percent interval) of the video frames separately. Table 2 shows that LSTR outperforms existing methods at every stage of action instances.

**HACS Segment.** LSTR achieves 82.6% on HACS Segment in term of mAP using Kinetics pretrained features. Note that HACS Segment is a new large-scale dataset with only a few previous results. LSTR outperforms existing methods RNN [28] (77.6%) by 5.0% and TRN [72] (78.9%) by 3.7%.

### 4.3.2 Action Anticipation

We extend the idea of LSTR to action anticipation for up to 2 seconds (*i.e.*, 8 steps in 4 FPS) into the future. Specifically, we concatenate another 8 learnable output tokens (with positional embedding) after the short-term memory in the LSTR decoder to produce the prediction results accordingly. Table 1c shows that LSTR significantly outperforms the state-of-the-art methods [72, 46] by 7.5% mAP on THUMOS and 2.1% cAP on TVSeries, using ActivityNet pretrained features.

### 4.4 Design Choices of Long- and Short-Term Memories

Table 3: **Results of LSTR using downsampled long-term memory** on THUMOS'14 in mAP (%). In particular, we use long-term memory of 512 seconds and short-term memory as 8 seconds.

| Temporal Stride | 1 | 2 | 4 | 8 | 16 | 32 | 64 | 128 |
|---|---|---|---|---|---|---|---|---|
| LSTR | 69.5 | 69.5 | 69.5 | 69.2 | 68.7 | 67.3 | 66.6 | 65.9 |

We experiment for design choices of long- and short-term memories. Unless noted otherwise, we use THUMOS'14, which contains various video lengths, and Kinetics pretrained features.

**Lengths of long- and short-term memories.** We first analyze the effect of different lengths of long-term $m_L$ and short-term $m_S$ memory. In particular, we test $m_S \in \{4, 8, 16\}$ seconds with $m_L$ starting from 0 second (no long-term memory). Note that we choose the max length (1024 seconds for THUMOS'14 and 256 seconds for HACS Segment) to cover lengths of 98% videos, and do not have proper datasets to test longer $m_L$. Fig. 3 shows that LSTR is beneficial from larger $m_L$ in most cases. In addition, when $m_L$ is short ($\leq 16$ in our cases), using larger $m_S$ obtains better results and when $m_L$ is sufficient ($\geq 32$ in our cases), increasing $m_S$ does not always guarantee better performance.

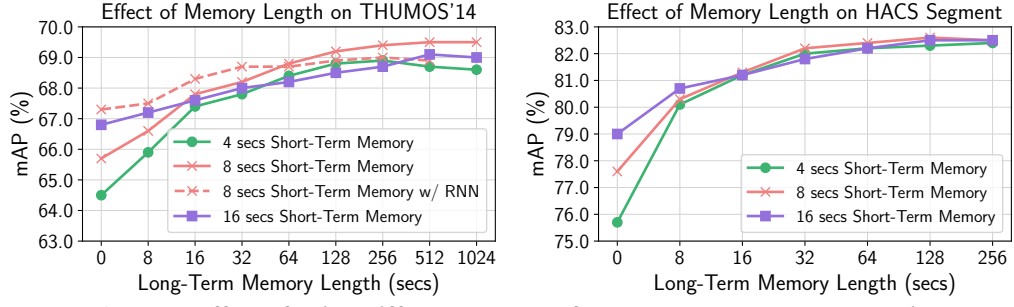

Figure 3: **Effect of using different lengths of long- and short-term memories**.

**Can we downsample long-term memory?** We implement LSTR with $m_S$ as 8 seconds and $m_L$ as 512 seconds, and test the effect of downsampling long-term memory. Table 3 shows the results that downsampling with strides smaller than 4 does not cause performance drop, but more aggressive

strides dramatically decrease the detection accuracy. Note that, when extracting frame features in 4 FPS, both LSTR encoder ($n_0 = 16$) and downsampling with stride 128 compress the long-term memory to 16 features, but LSTR achieves much better performance (69.5% vs. 65.9% in mAP). This demonstrates the effectiveness of our "adaptive compression" compared to heuristics downsampling.

**Can we compensate reduced memory length with RNN?** We note that LSTR's performance notably decreases when it can only access very limited memory (*e.g.*, $m_L + m_S \leq 16$ seconds). Here we test if RNN can compensate LSTR's reduced memory or even fully replace the LSTR encoder. We implement LSTR using $m_S = 8$ seconds with an extra Gated Recurrent Unit (GRU) [10] (its architecture is visualized in the Supplementary Material) to capture all history outside the long- and short-term memories. The dashed line in Fig. 3 shows the results. Plugging-in RNNs indeed improves the performance when $m_L$ is small, but when $m_L$ is large ($\geq 64$ seconds), it does not improve the accuracy anymore. Note that RNNs are not used in any other experiments in this paper.

### 4.5 Design Choices of LSTR

Table 4: **Results of different designs of the LSTR encoder and decoder**. The length of short-term memory is set to 8 seconds. "TR" denotes Transformer. The last row is our proposed LSTR design.

| LSTR Encoder | LSTR Decoder | Length of Long-Term Memory $m_L$ (secs) | | | | | | | |
|:---:|:---:|:---:|:---:|:---:|:---:|:---:|:---:|:---:|:---:|
| | | 8 | 16 | 32 | 64 | 128 | 256 | 512 | 1024 |
| *N/A* | TR Encoder | 65.7 | 66.8 | 67.1 | 67.2 | 67.3 | 66.8 | 66.5 | 66.2 |
| *N/A* | TR Decoder | 66.5 | 67.3 | 67.7 | 68.1 | 68.3 | 67.9 | 67.0 | 66.5 |
| TR Encoder | TR Decoder | 65.9 | 66.4 | 66.7 | 67.4 | 67.5 | 67.2 | 67.0 | 66.6 |
| Projection Layer | TR Decoder | 66.2 | 67.1 | 67.4 | 67.7 | 67.5 | 67.2 | 66.9 | 66.8 |
| TR Decoder | TR Decoder | 66.1 | 67.1 | 67.4 | 68.0 | 68.5 | 68.6 | 68.7 | 68.7 |
| TR Decoder + TR Encoder | TR Decoder | 66.2 | 67.3 | 67.6 | 68.4 | 68.6 | 68.8 | 68.9 | 69.0 |
| TR Decoder + TR Decoder | *N/A* | 64.0 | 64.7 | 65.9 | 66.1 | 66.5 | 66.2 | 65.4 | 65.2 |
| **TR Decoder + TR Decoder** | **TR Decoder** | **66.6** | **67.8** | **68.2** | **68.8** | **69.2** | **69.4** | **69.5** | **69.5** |

We continue to explore the design trade-offs of LSTR. Unless noted otherwise, we use short-term memory of 8 seconds, long-term memory of 512 seconds, and Kinetics pretrained features.

**Number of layers and tokens.** First, we assess using different numbers of token embeddings (*i.e.*, $n_0$ and $n_1$) in LSTR encoder. Fig 4 (left) shows that LSTR is quite robust to different choices (the best and worst performance gap is only about 1.5%), but using $n_0 = 16$ and $n_1 = 32$ gets highest accuracy. Second, we experiment for the effect of using different numbers of Transformer decoder units (*i.e.*, $\ell_{enc}$ and $\ell_{dec}$). As shown in Fig 4 (right), LSTR does not need a large model to get the best performance, and in practice, using more layers can cause overfitting.

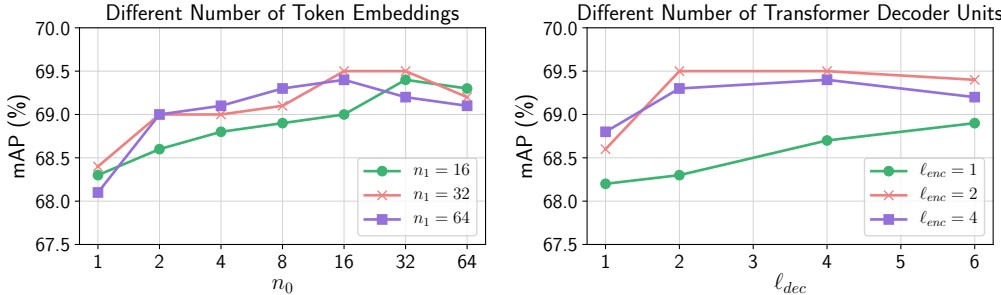

Figure 4: **Left**: Results of different number of token embeddings for our two-stage memory compression. **Right**: Results of different number of Transformer decoder units for $\ell_{enc}$ and $\ell_{dec}$.

**Can we unify the temporal modeling using only self-attention models?** We test if long-term $\mathbf{M}_L$ and short-term $\mathbf{M}_S$ memory can be learned as a whole using self-attention models. Specifically, we concatenate $\mathbf{M}_L$ and $\mathbf{M}_S$ and feed them into a standard Transformer Encoder [61] with a similar model size to LSTR. Table 4 (row 8 vs. row 1) shows that LSTR achieves better performance especially when $m_L$ is large (*e.g.*, 69.5% vs. 66.5% when $m_L = 512$ and 66.6% vs. 65.7% when $m_L = 8$). This shows the advantage of LSTR for temporal modeling on long- and short-term context.

**Can we remove the LSTR encoder?** We explore this by directly feeding $\mathbf{M}_L$ into the LSTR decoder, and using $\mathbf{M}_S$ as tokens to reference useful information. Table 4 shows that LSTR outperforms this

baseline (row 8 vs. row 2), especially when $m_L$ is large. We also compare it with self-attention models (row 1) and observe that although neither of them can effectively model prolonged memory, this baseline outperforms the Transformer Encoder. This also demonstrates the effectiveness of our idea of using short-term memory to query related context from long-range context.

**Can the LSTR encoder learn effectively using self-attention?** To evaluate the "bottleneck" design with cross-attention in LSTR encoder, we try modeling $\mathbf{M}_L$ using standard Transformer Encoder units [61]. Note that this still captures $\mathbf{M}_L$ and $\mathbf{M}_S$ with the similar workflow of LSTR, but does not compress and encode $\mathbf{M}_L$ using learnable tokens. Table 4 (row 8 vs. row 3) shows that LSTR outperforms this baseline with all $m_L$ settings. In addition, the performance of this baseline decreases when $m_L$ gets larger, which suggests the superior ability of LSTR for modeling long-range patterns.

**How to design the memory compression for the LSTR encoder?** First, we use a projection layer, consisting of a learnable matrix of size $n_0 \times m_L$ followed by MLP layers, to compress the long-term memory along the temporal dimension. Table 4 shows that using this simple projection layer (row 4) slightly outperforms the model without long-term memory (row 1), but is worse than attention-based compression methods. Second, we evaluate the one-stage design with $\ell_{enc} + 1$ Transformer decoder units. Table 4 (row 8 vs. row 5) shows that two-stage compression is stably better than one-stage, and their performance gap gets larger when using larger $m_L$ (0.5% when $m_L = 8$ and 0.8% when $m_L = 512$). Third, we compare cross-attention and self-attention for two-stage compression by replacing the second Transformer decoder with Transformer encoder. LSTR stably outperforms this baseline (row 6) by about 0.5% in mAP. However, its performance is still better than models with one-stage compression of about 1.3% (row 6 vs. row 3) and 0.3% (row 6 vs. row 5) on average.

**Can we remove the LSTR decoder?** We remove the LSTR decoder to evaluate its contribution. Specifically, we feed the entire memory to LSTR encoder and attach a multi-layer (MLP) classifier on its output tokens embeddings. Similar to the above experiments, we increase the model size to ensure a fair comparison. Table 4 shows that LSTR outperforms this baseline (row 7) by about 4% on large $m_L$ (*e.g.*, 512 and 1024) and about 2.5% on relative small $m_L$ (*e.g.*, 8 and 16).

Table 5: **Results of LSTR using different memory integration methods** in mAP (%). Our proposed integration method using cross-attention stably outperforms the heuristic methods.

| Memory Integration Methods | Length of Long-Term Memory $m_L$ (secs) | | | | | | | |
|---|---|---|---|---|---|---|---|---|
| | 8 | 16 | 32 | 64 | 128 | 256 | 512 | 1024 |
| Average Pooling | 66.1 | 67.0 | 67.3 | 67.5 | 68.2 | 68.4 | 68.6 | 68.6 |
| Concatenation | 65.9 | 67.2 | 67.5 | 67.7 | 68.4 | 68.5 | 68.7 | 68.6 |
| **Cross-Attention** (ours) | **66.6** | **67.8** | **68.2** | **68.8** | **69.2** | **69.4** | **69.5** | **69.5** |

**Cross-attention vs. heuristics for integrating long- and short-term memories.** We explore integrating the long- and short-term memories by using average pooling and concatenation. Specifically, the encoded long-term features of size $n_1 \times C$ is converted to a vector of $C$ elements by channel-wise averaging, and the short-term memory of size $m_S \times C$ is encoded by $\ell_{dec}$ Transformer encoder units. Then, each slot of the short-term features is either averaged or concatenated with the long-term feature vector for action classification. Note that these models still benefit from LSTR's effectiveness for long-term modeling. Table 5 shows that using average pooling and concatenation obtain comparable results, but LSTR with cross-attention stably outperforms these baselines.

## 4.6 Runtime

Table 6: **Runtime of LSTR with different design choices**. The last row is our proposed LSTR design.

| LSTR Encoder | LSTR Decoder | Frames Per Second (FPS) | | | |
|---|---|---|---|---|---|
| | | OptFlow Computation | RGB Feature Extraction | OptFlow Feature Extraction | LSTR |
| *N/A* | TR Encoder | | | | 43.2 |
| TR Encoder | TR Decoder | 8.1 | 70.5 | 14.6 | 50.2 |
| TR Decoder | TR Decoder | | | | 59.5 |
| **TR Decoder + TR Decoder** | **TR Decoder** | | | | 91.6 |

We report LSTR's runtime in frames per second (FPS) on a system with a single V100 GPU, and use the videos from THUMOS'14 dataset. The results are shown in Table 6.

We start by comparing the runtime between LSTR's different design choices without considering the pre-processing (*e.g.*, feature extraction). First, LSTR runs at 91.6 FPS using our two-stage memory compression (row 4), whereas using the one-stage design runs at a slower 59.5 FPS (row 3). Our two-stage design is more efficient because it does not need to reference information from the long-term memory multiple times, and can be further accelerated during online inference (see Sec. 3.6). Second, we test the LSTR encoder using self-attention mechanisms (row 2). This design does not compress the long-term memory, thus increasing the computational cost of both the LSTR encoder and decoder, leading to a slower speed of 50.2 FPS. Third, we test the standard Transformer Encoder [61] (row 1), whose runtime speed, 43.2 FPS, is about $2\times$ slower than LSTR.

We also compare LSTR with state-of-the-art recurrent models. As we are not aware of any prior work that reports their runtime, we test TRN [72] using their official open-source code [2]. The result shows that TRN runs at 123.3 FPS, which is faster than LSTR. This is because recurrent models abstract the entire history as a compact representation but LSTR needs to process much more information. On the other hand, LSTR achieves much higher performance, outperforming TRN by about $7.5\%$ in mAP on THUMOS'14 and about $4.5\%$ in cAP on TVSeries.

For end-to-end online inference, we follow the state-of-the-art methods [72, 19, 25] and build LSTR on two-stream features [62]. LSTR together with pre-processing techniques run at 4.6 FPS. Table 6 shows that the speed bottleneck is the motion feature extraction — it accounts for about 90% of the total runtime including the optical flow computation with DenseFlow [63]. One can improve the efficiency largely by using real-time optical flow extractors (*e.g.*, PWC-Net [55]) or using only visual features extracted by a light-weight backbone (*e.g.*, MobileNet [29] and FBNet [67]).

## 4.7 Error Analysis

Table 7: **Action classes with highest and lowest performance** on THUMOS'14.

| Action Classes | HammerThrow | PoleVault | LongJump | Diving | BaseballPitch | FrisbeeCatch | Billiards | CricketShot |
|---|---|---|---|---|---|---|---|---|
| AP (%) | 92.8 | 89.7 | 86.9 | 86.7 | 55.4 | 49.4 | 39.8 | 38.6 |

Figure 5: **Failure cases** on THUMOS'14. Action classes from left to right are "BaseballPitch", "FrisbeeCatch", "Billiards", and "CricketShot". Red circle indicates where the action is happening.

In Table 7, we list the action classes from THUMOS'14 where LSTR gets the highest (color green) and the lowest (color red) per-frame APs. In Fig. 5, we illustrate four sample frames with incorrect predictions. More visualizations are included in the Supplementary Material. We observe that LSTR sees a decrease in detection accuracy when the action incurs only tiny motion or the subject is very far away from the camera, but excels at recognizing actions with long temporal span and multiple stages, such as "PoleVault" and "Long Jump". This suggests we may explore extending the temporal modeling capability of LSTR to both spatial and temporal domains.

## 5 Conclusion

We present LSTR which captures both long- and short-term correlations in past observations of a time series by compressing long-term memory into encoded latent features and referencing related temporal context from them with short-term memory. This demonstrates the importance of separately modeling long- and short-term information and then integrating them for online inference tasks. Experiments on multiple datasets and ablation studies validate the effectiveness and efficiency of the LSTR design in dealing with prolonged video sequences. However, we note that LSTR is operating only on the temporal dimension. An end-to-end video understanding system requires simultaneous spatial and temporal modeling for optimal results. Therefore extending the idea of LSTR to spatio-temporal modeling remains an open yet challenging problem.

## 6 Acknowledgments and Disclosure of Funding

We thank the anonymous reviewers for their helpful suggestions. This work was funded by Amazon.

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
