# Long Short-Term Transformer for Online Action Detection Supplementary Material

**Mingze Xu**     **Yuanjun Xiong**     **Hao Chen**     **Xinyu Li**
**Wei Xia**     **Zhuowen Tu**     **Stefano Soatto**
Amazon/AWS AI
{xumingze,yuanjx,hxen,xxnl,wxia,ztu,soattos}@amazon.com

## A    Can we compensate reduced memory length with RNN? Cont'd

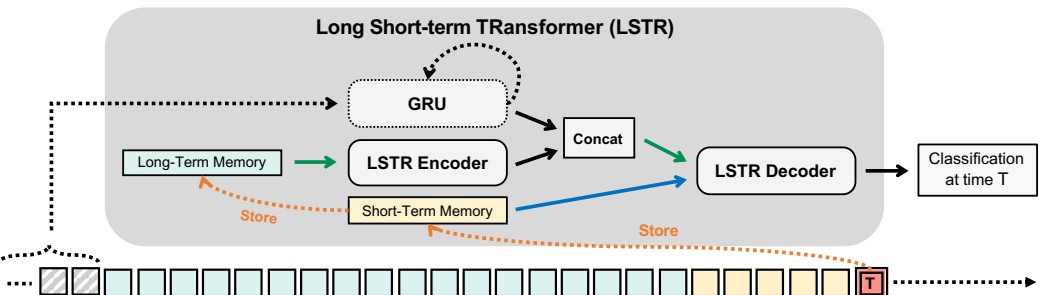

Figure 1: **Overview of the architecture of using LSTR with an extra GRU.**

To better understand the design choice in Sec. 4.4, we show its overall structure in Fig. 1. Specifically, in addition to the long- and short-term memories, we use an extra GRU to capture all the history "*outside*" the long- and short-term memories as a compact representation, $\mathbf{g} \in \mathbb{R}^{1 \times C}$. We then concatenate the outputs of the LSTR encoder and the GRU as more comprehensive temporal features of size $(n_1 + 1) \times C$, and feed them into the LSTR decoder as input tokens.

## B    Action Anticipation Cont'd

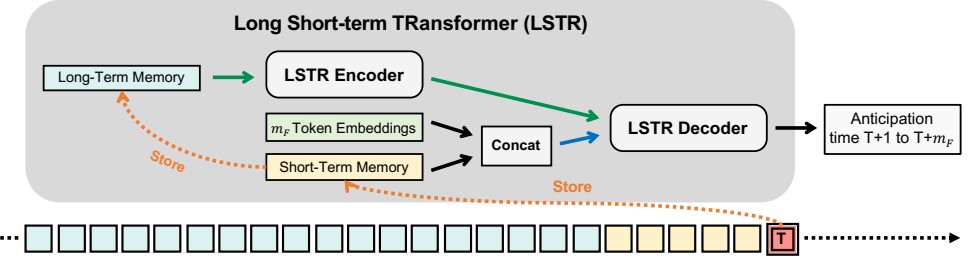

Figure 2: **Overview of the architecture of using LSTR for action anticipation.**

In Sec. 4.3.2, we extended the idea of LSTR to action anticipation for up to 2 seconds into the future, and compared it with state-of-the-art methods in Table 1c. Here we provide more details about its structure. As shown in Fig. 2, we concatenate $m_F$ token embeddings after the short-term memory as $m_S + m_F$ output tokens into the LSTR decoder. These $m_F$ "anticipation tokens" are added with the positional embedding and directional attention mask together with the short-term memory. Then the outputs of the $m_F$ tokens make predictions for the next $m_F$ steps accordingly. As LSTR's performance is evaluated in 4 FPS (see Sec. 4.2), $m_F$ is set to 8 for action anticipation of 2 seconds.

35th Conference on Neural Information Processing Systems (NeurIPS 2021).

# C  Qualitative Results

Fig. 3a shows some qualitative results. We can see that, in most cases, LSTR can quickly recognize the actions and make relatively consistent predictions for each action instance. Two typical failure cases are shown in Fig. 3b. The top sample contains the "Billiards"action that incurs only tiny motion. As discussed in Sec. 4.7, LSTR's detection accuracy is observed to decrease on this kind of actions. The bottom sample is challenging — the "Open door" action occurs behind the female reporter and is barely visible. Red circle indicates where the action is happening in each frame.

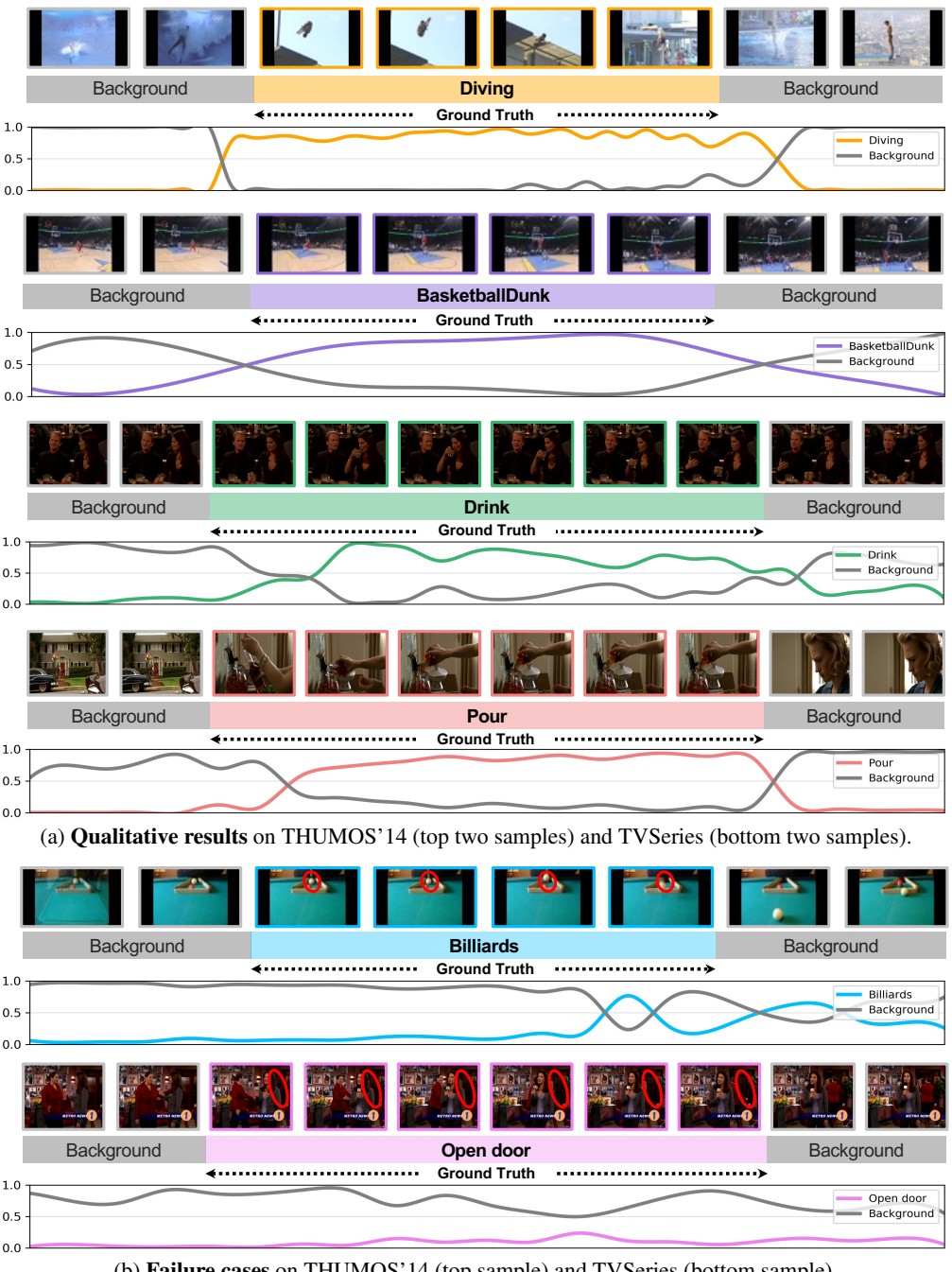

(a) **Qualitative results** on THUMOS'14 (top two samples) and TVSeries (bottom two samples).

(b) **Failure cases** on THUMOS'14 (top sample) and TVSeries (bottom sample).

Figure 3: **Qualitative results and failure cases of LSTR**. The curves indicate the predicted scores of the ground truth and "background" classes. (Best viewed in color.)