# OpenReview forum: "Long Short-Term Transformer for Online Action Detection"
_NeurIPS.cc/2021/Conference — NeurIPS 2021 Spotlight_

### Official Review · Reviewer_eqy6 · 2021-07-14

**Rating:** 7
**Confidence:** 5

**Summary:**

This paper presents a novel long video model named long short term transformer, with online action detection as the study case. The proposed transformer uses the encoder to model long-term historical information and uses the decoder to model the short-term information i.e. a few frames right before the current time. Experiments have been done on three standard benchmarks (THUMOS, TV, HACS) and the performance gains are clear compared to SOTA. Detailed analysis has been conducted as well.

**Limitations And Societal Impact:**

The authors have described the limitations of the model. As described in Sec 4.6, this model is not robust for tiny motion and a lower performance when the subject is very far away from the camera.

**Main Review:**

Strength:
1. The paper is well written and organized.
2. The problem of online action detection is important in real-world applications and the proposed idea is interesting in the sense of being capable of dealing with very long video by decoupling long and short memories.
3. This model outperforms all of the sota methods on three benchmark datasets with clear gains, confirming the effectiveness of the proposed approach. Notably, the paper adopted LFB and tried to reproduce it on the THUMOS dataset for comparison.
4. The ablation experiments are extensive and can demonstrate the advantages of each component of the LSTR architecture.

Weakness:
1. The time complexity analysis in Sec 3.3 and Sec 3.6 is concrete and direct, however, there are no related experiments to evaluate the reduced time complexity in the paper. Although the runtime is shown in the Appendix, it is highly recommended to include the results in the experiment section since one important novelty of this model is its efficiency (as claimed in line 9 and line 40). Also, it is better to compare the efficiency of the memory compression with the single-stage design.
2. The idea of compressing historical information and decoupling long and near term information is not new for temporal sequence modeling. “Compressive transformers for long-range sequence modelling. ICLR 2020.” has similar motivation. I do acknowledge there are significant efforts in adapting the similar idea to online action detection, such a specific topic. But it's worthwhile mentioning and discussing these NLP works about dealing with long sequences.
3. It is nice that the paper compares with LFB [58] on THUMOS’14 because the non-local block can also aggregate features based on the attention operator. But in addition to TH’14, it will better convince me if there are comparisons with LFB on TVseries and HACS datasets as well.
4. This work only focuses on online action detection - how about offline temporal localization? In other words, I wonder if this kind of long short term modeling might also be useful for offline scenarios such as modeling long-term dependencies of sub-events in a long event while at the same time capturing fine-grained, localized information from short yet key moments. If space allowed, it would be interesting to discuss for future work.


**Time Spent Reviewing:**

2.5

---

> ### Author Response · Authors · 2021-08-10
> **Response to Reviewer eqy6**
>
> Thank you for your review. We are encouraged that you find our idea interesting and effective, and our paper well-written and organized. We address your concerns below.
>
> **Q1. “The time complexity analysis in Sec 3.3 and Sec 3.6 is ... compression with the single-stage design.”**
>
> **Answer:** Thanks for this good suggestion. We will move the runtime analysis from Appendix to the main paper in the revision. The runtime of the single-stage memory compression implemented with both TR Encoder and TR Decoder are included in Appendix’s Table 1 (row 2 and 3, respectively). The results are copied to the following table. We observe that our two-stage design runs much faster than the single-stage design, which validates our time complexity analysis in Sec 3.6.
>
> | Memory Compression Method | Runtime of Action Detection Model |
> | :---- | :----: |
> | Single-Stage Design w/ TR Encoder | 50.2 FPS |
> | Single-Stage Design w/ TR Decoder | 59.5 FPS |
> | Two-Stage Design (TR Decoder + TR Decoder) | 83.9 FPS |
>
> **Q2. “The idea of compressing historical information ... dealing with long sequences.”**
>
> **Answer:** Thank you for introducing related work such as [1]. We believe LSTR is distinct from [1] in several key aspects. First, LSTR aims at modeling both the long-term memory (providing long-range contextual information for the history) and the short-term memory (providing informative prompt information), whereas Compressive Transformers [1] aim to compress long-range past activations into coarser memories. Second, LSTR and Compressive Transformers [1] are motivated and implemented differently. Compressive Transformers [1] focuses primarily on memory compression, which uses standard operations, such as pooling and convolution, to compress samples older than the allowed inference window size to a reduced temporal resolution. On the other hand, LSTR’s target scope is beyond memory compression. It aims at modeling comprehensive temporal data by distilling/exploiting latent representation (as representative agents) from long-range sequence via a two-stage network. Third, LSTR utilizes the transformer decoder unit to distill long-term memory of an arbitrary size into a fixed length representation while Compressive Transformer [1] performs fixed-ratio temporal downsampling. This could make LSTR more applicable to modeling variable length history information.
>
> We will update [1] to the “Related Work” section accordingly and review more related work of NLP in the reversion.
>
> [1] Compressive transformers for long-range sequence modeling. ICLR 2020.
>
> **Q3. “It is nice that the paper compares with LFB ... TVseries and HACS datasets as well.”**
>
> **Answer:** The results of LFB on TVSeries were reported in paper’s Table 1. As suggested, we also evaluated the performance of LFB on HACS using the same experimental settings. The result shows that LFB achieves 80.1% mAP on HACS, and LSTR (82.6% mAP) outperforms this strong baseline by 2.5%. We will add these results into the main paper.
>
> **Q4. “This work only focuses on online action detection ... to discuss for future work.”**
>
> **Answer:** Thanks for this good suggestion. In this paper we focus on online action detection, since it is a typical problem that requires jointly modeling the long- and short-term context information. We believe that frames observed recently provide precise information about the ongoing action instance, while long-range historical observations offer contextual references for actions that are potentially happening right now. As suggested, we will discuss the possibility of applying LSTR to other action understanding tasks, such as offline temporal localization, as future work in the revision.

---

> > ### Comment · Reviewer_eqy6 · 2021-09-01
> > **Final review**
> >
> > Thank you authors for a detailed response. I have read the reviews from other reviewers too. My concerns have been adequately addressed by the rebuttal. I am going to keep my previous rating and recommend an acceptance for this paper.

---

### Official Review · Reviewer_eKmB · 2021-07-18

**Rating:** 7
**Confidence:** 4

**Summary:**

The paper addresses an interesting and practical problem in videos – online detection of actions by examining only frames available till the present time and without any access to future frames. The problem is not only challenging but has practical significance too as online action detection is important in autonomous driving or surveillance scenarios and handling this with fast and light memory-weight training and inference enables one to address the problem real-time. The solution approach is based on the intuition that recent frames are more instrumental about the ongoing action while frames from extended past offer potential context to the present action. While differential exploitation of recent and extended past can definitely help (compared to using frames only from the recent past), it comes with extra computation and memory overhead. Based on the recently popular transformer philosophy, the authors came up with an encoder-decoder architecture with short and long term memories for recent and extended memories respectively. While related works have explored long and short term memories, the integration of the two outputs from them is not explicit and rather ad-hoc (pooling, concatenation etc.). The proposed work, on the other hand, made intelligent use of transformers with a two-stage memory compression design based on transformer decoders. Such a design is shown to be not only effective but also both memory and compute efficient. The experiments are well-designed. Extensive comparison with SOTA approaches and well-thought ablations show its superiority and robustness of design choices. The writing is crisp and adequate with insightful experimental analysis. The figures are also self-explanatory with descriptive captions. The related works are also covered good.

**Limitations And Societal Impact:**

It is described adequately in section 4.6 and possible future exploration of the temporal modeling capability of LSTR to both the spatial and the temporal domains is planned.

**Main Review:**

[Strengths]:
1. The paper addresses a very practical problem and proposes an efficient transformer based architecture for the purpose.
2. The experiments are well-designed and executed showing the superiority of the proposed approach.
3. The paper is well-written, the figures are well-illustrated and related works are well documented.
4. The long and short term memory with two-stage memory compression design based on transformer decoders for online action detection is novel and effective.

[Weaknesses]:

Though I am writing these under the heading ‘weaknesses’, they are mostly my queries for a better understanding.
1. Line 69-71 mentions about existing non-explicit integration of long and short term features using simple pool or concat operations. An ablation experiment with such heuristics for the integration would have been good.
2. Though online action detection is a very good usecase, is it possible to show the applicability of LSTR's superior modeling of long videos in other video applications?
3. Figure 1 (since the figure is said as best viewed in color): Does the different arrow colors mean anything?

In short, the paper seems to have addressed a practical problem with well-supported experiments and analysis and is also well written. Thus, I am in favor of accepting it at the moment.

**Time Spent Reviewing:**

7

---

> ### Author Response · Authors · 2021-08-10
> **Response to Reviewer eKmB**
>
> Thank you for your review. We are encouraged that you find our two-stage memory compression design novel, and the experimental analysis thorough and insightful. We address your concerns below.
>
> **Q1. “Line 69-71 mentions about existing non-explicit integration ... would have been good.”**
>
> **Answer:** Thanks for suggesting this good ablation study. As suggested, we test to integrate the long- and short-term memories with average pooling and concatenation. Specifically, the encoded long-term memory of size $n_1 \times C$ is first abstracted to a feature vector of size $C$ by averaging along the temporal dimension. The short-term memory of size $m_S \times C$ is encoded by a multi-layer Transformer Encoder to fully capture the local context. Then, each slot of the encoded short-term features is averaged (or concatenated) with the long-term feature vector for making classification. Note that these baselines still benefit from LSTR's effectiveness for long-term modeling with comparable model size, and the main difference is that they use simple integration methods instead of cross-attention. Other experimental settings (e.g., hyper-parameters) are identical to Sec 4.3.
>
> | Memory Integration Method | 8s     | 16s    | 32s    | 64s    | 128s   | 256s   | 512s   |
> | :---- | :----: | :----: | :----: | :----: | :----: | :----: | :----: |
> | Average Pooling       | 66.1 | 67.0 | 67.3 | 67.5 | 68.2 | 68.4 | 68.6 |
> | Concatenation          | 65.9 | 67.2 | 67.5 | 67.7 | 68.4 | 68.5 | 68.7 |
> | Cross-Attention         | 66.6 | 67.8 | 68.2 | 68.8 | 69.2 | 69.4 | 69.5 |
>
> The evaluation results on THUMOS are shown in the following table. We observe that using average pooling (row 1) and concatenation (row 2) obtain comparable results, but LSTR that uses cross-attention for memory integration (row 3) stably outperforms these baselines. We will add this ablation study in the revision.
>
> **Q2. “Though online action detection is ... long videos in other video applications?”**
>
> **Answer:** Thanks for this good idea. We agree that LSTR has the potential to be a general framework for modeling long-range context. Here we focus on online action detection because it is a typical problem that requires jointly modeling the long- and short-term temporal information. Specifically, frames observed recently provide precise information about the ongoing action instance, while long-range historical observations offer contextual references for actions that are potentially happening right now. We agree that exploration in other video understanding tasks, such as action recognition and multiple object tracking, could be an interesting direction for future work. As suggested, we will discuss the applicability of LSTR to these applications as future work in the revision.
>
> **Q3. “Figure 1 (since the figure is said as best viewed in color) ... colors mean anything?”**
>
> **Answer:** The dark green denote input tokens and the dark blue arrows denote output tokens, which correspond to the same meaning in Fig 2. The orange, dashed arrows indicate the data flow (e.g., storing features in the long- and short-term memories). We will clarify this in the revision.

---

> > ### Comment · Reviewer_eKmB · 2021-08-22
> > **Final thoughts**
> >
> > Had a read of the response as well as the reviews from my fellow reviewers. The authors have adequately addressed my queries. Especially the ablation of LSTR with average pooling and concatenation shows its superiority over traditional ad-hoc integration methods. The authors have also addressed queries from my fellow reviewers with additional experiments as and when needed. So, I am going with an accept for the paper and keeping my previous rating as is.

---

### Official Review · Reviewer_4xKN · 2021-07-19

**Rating:** 6
**Confidence:** 4

**Summary:**

This work proposes a two-stage processing of video frames by Transformer decoders for video classification.  In the first stage a set of learnt queries “decode” a sequence of fixed-number of earlier video frames. Then in the second stage, features from recent frames act as queries and further “decode” the vectors output in the first stage. At the end of the second stage, final classification is carried out.

The main contribution is this two-stage design of the video classification model.
The fixed-length “long-term” context of video frames is shown to be useful for “online” frame classification (i.e., w/o access to the future frames).

This work presents a set of experiments for different design choices, which is also informative.

The key experiments are online video classification on — THUMOS’14, TVSeries, and HACS datasets, where the proposed model outperforms the baselines reported in this work.

**Ethical Concerns:**

No.

**Limitations And Societal Impact:**

No.

1. Potential applications in online video classification (e.g., surveillance).


**Main Review:**

### Originality
The use of slot-based video features for long-term memory and its combination with short-term features using attention-based non-local blocks was introduced in long-term feature banks [28]. However, this work proposes a specific architecture using Transformer Decoders which is novel.
In the online action “detection” space, this method achieves strong results.
&nbsp;

### Quality
The design choices are validated through a good set of experiments. However, key weaknesses include: (i) use of non-standard datasets (however, these datasets seem to be standard for “online action detection”), and (ii) unclear if baseline methods were matched in capacity/modality etc. (see detailed comments below).
&nbsp;

### Clarity
The writing is concise and easy to understand. However, there are a few potential errors (e.g., (3) in “clarifications/typos” below).
Significance
Given that the using slot-based video features for long-term context, and absence of results on more popular/standard datasets might limit the impact of this work.
&nbsp;

### Strengths
The proposed method is simple/easy to understand and implement.
A number of experiments validate the various design choices.
Strong results are achieved by the proposed model on a number of popular “online action detection” benchmarks.
&nbsp;

### Weakness
1. The proposed model has a fixed context length (although this can be large).  This means that information from frames older than a certain length ($m_L + m_s$) do not influence the current decision. In LSTMs, for example, this context-length is continuous and not fixed (at least in theory). Hence, the model design does not accurately represent the name of the proposed model “LSTR”.

2. On L268, an experiment is performed with recurrent models to compensate for this limited context length. However, it is found that this has no benefit over long LSTR memory sizes. This could be a limitation of the tasks in the experiment / difficulty in training the RNNs.


3. A second limitation is that experimental comparisons against baselines are provided under non-standard settings. For example, it is not clear whether the baselines (e.g., LFB [58]) have a similar number of model parameters, model depth/backbone, modality ( i.e., two-stream (RGB + flow) or not), and inference time.  If the authors can add this information to the tables 1(a,b) and 2, it would help put the proposed method in context.


4. Further to the point above, a main reference model is the LFB [58], as it similarly has slot-based long-term memory.  However, the proposed work does not compare to LFB in the settings explored in the original LFB paper (e.g., Charades / EPIC-Kitchens – i.e. action recognition without localisation).


5. The comparison against “standard Transformer Encoder units” is provided on L297-L303. However, no comparisons are provided against standard Transformer **decoder** units, which are what the proposed model is based on. Comparison against a stack of Transformer decoder units (with learned query vectors, very much like the model proposed in [28]), without the late-stage fusion with recent features will be informative. This is a key ablation.


6. While the use of Transformers with “two-stage” video memory is novel, it is not very different in principle from the LFB [58] model, which also uses attention-based non-local blocks for fusing short-term and long-term features.   This might limit the impact of the proposed work.


7. THUMOS’14, and TVSeries are relatively small datasets (although popularly used in “online action detection” works), as compared to more “standard” datasets, like EPIC-kitches, Charades, AVA etc., which might further limit the impact of the proposed work. Authors are encouraged to adapt their method to the above datasets for greater impact.
&nbsp;

### Clarifications / Typos:
1. L79:  “most of these work only performs …” → these works … perform.


2. L92: what is ‘t’ in (txC) dimensional temporal sequence? Did you mean \tau instead?


3. L159: Unclear how $O(n_0 * (m_L +C))$ is “sub-linear w.r.t $m_L$”; clearly it grows linearly with $m_L$. Please clarify.


4. L166: Saying $m_S$ are “output tokens” can be improved, by using the term “queries” instead.


5. L173:  Not having to do temporal unrolling / BPTT has more to do with the fact that the proposed architecture drops features older than $m_L$ time-steps (due to the FIFO queue), than using Transformers.

6. Sec. 3.5:  Training for all frames in the short-term memory implies that the context length is shorter for the earlier $m_S - 1$ frames – this may be beneficial due to regularization, however, this point should be discussed.


**Time Spent Reviewing:**

4

---

> ### Author Response · Authors · 2021-08-10
> **Response to Reviewer 4xKN**
>
> Thank you for your insightful comments. We appreciate that you find our two-stage memory design novel, and our paper easy to understand. We address your concerns below.
>
> **Q1. “The proposed model has a fixed context ... the name of the proposed model “LSTR”.”**
>
> **Answer:** We name our model as LSTR because it introduces the long- and short-term memory design to jointly capture the temporal information for online action detection problem. In our experiments, LSTR does not use infinite memory size.  However, LSTR proposes a new perspective to directly store and reference the related history information, which effectively mitigates the problem of quick forgetting [1] in modeling long-range context.
>
> [1] Neural Turing Machines. Graves et al. Neural and Evolutionary Computing, 2014.
>
> **Q2. “On L268, an experiment is performed with ... difficulty in training the RNNs.”**
>
> **Answer:** Thanks for this insightful comment. It is possible that samples in existing online action detection datasets does not benefit from extra long-range context (e.g., over 500 seconds/8 minutes). On the other hand, we agree that a more effective combination of recurrent networks and LSTR for generalized video understanding could be a very interesting direction to explore as future work.
>
> **Q3. “A second limitation is that experimental ... proposed method in context.” and Q6. “While the use of Transformers with “two-stage” video memory is novel ... the proposed work.”**
>
> **Answer:** LFB focuses on modeling long-term features, whereas LSTR targets jointly modeling the long- and short-term memories as integrated context information. To ensure fair comparison, we used the same pre-trained feature extractors and input modalities for LFB and LSTR. We also increased the number of non-local blocks for LFB to make sure its model size is comparable to LSTR. More details are shown in the following table, and we will clarify this in the revision.
>
> | Model | #Parameters | Modality | RGB Backbone | Flow Backbone |
> | :---- | :----: | :----: | :----: | :----: |
> | LFB  | 51M | RGB + Flow | ResNet-50 | BN-Inception |
> | LSTR | 58M | RGB + Flow | ResNet-50 | BN-Inception |
>
> **Q4. “Further to the point above, a main reference ... without localisation).” and Q7. “THUMOS’14, and TVSeries are relatively small ... above datasets for greater impact.”**
>
> **Answer:** In this paper, we also evaluated LSTR on HACS Segment dataset, which contains 35,300 untrimmed videos over 200 human action classes. The results show that LSTR can achieve convincing results on this large-scale benchmark. In this paper, we focus on online action detection because it is a typical problem that requires jointly modeling the long- and short-term temporal information. However, we agree that LSTR can have larger impact if it can be effectively applied on other video tasks. We plan to explore on this research direction in the future.
>
> **Q5. “The comparison against “standard Transformer Encoder units” ... This is a key ablation.”**
>
> **Answer:** We reported the performance of using only Transformer Decoder units in paper’s Table 4 (row 6). For this baseline, stacked Transformer Decoder units with learnable token embeddings are used to jointly learn the current and historical observations. We found that LSTR outperforms this baseline by about 4.0% on large $m_L$ (e.g., 512 and 1024) and about 2.5% on relative small $m_L$ (e.g., 8 and 16). We will clarify this in the revision.
>
> **Q8. “Clarifications / Typos”**
>
> **Answer:** Thanks for these helpful suggestions. We will incorporate all minor suggestions (e.g., gramma issues and typos) in the revision.
>
> * **Q8.2 “L92: what is ‘t’ in (txC) ... Did you mean \tau instead?”**
>     * **Answer:** Thanks and yes, it is $\tau$. Will fix it.
> * **Q8.3 “L159: Unclear how O(n0∗(mL+C)) ... linearly with mL. Please clarify.”**
>     * **Answer:** Thanks for pointing this out. Yes, we agree that the time complexity of $O(n_0 \times (m_L + C))$ is linear with $m_L$. To be more clear, the amortized time complexity of computing the attention weights can be reduced from $O(n_0 \times m_L \times C)$ to $O(n_0 \times (m_L + C))$. Although the time complexity of the cross-attention operation is still $O(n_0 \times m_L \times C)$ due to the inevitable operation of weighted sum, considering $C$ is usually larger than 1024, this is still a considerable reduction of runtime cost. The runtime analysis in Appendix’s Table 1 validates our analysis here. We will further clarify this analysis in the revision.
> * **Q8.6 “Sec. 3.5: Training for all frames in the ... this point should be discussed.”**
>     * **Answer:** Thanks for pointing this out. Yes, we believe that training LSTR with various short-term memory lengths for each frame can serve as data augmentation. In that way, LSTR will not overfit to the pre-defined memory length (e.g., 8s). Using the directional attention mask is a way to efficiently and effectively implement this idea. Empirically, we found this data augmentation indeed significantly facilities the training of LSTR --- the model experienced severe overfitting problems without this design. Besides this, the directional attention mask also ensures that there is no violation of the online setting (i.e., prediction of the present cannot see the future) in the short-term memory. So that, we can efficiently provide supervision on every frame in the short-term memory (L179-L183). We will clarify this in the revision.

---

> > ### Comment · Reviewer_4xKN · 2021-08-31
> > **Final Review**
> >
> > Thank the authors for their response to the concerns raised in the review.  The authors have not addressed the concerns regarding (i) comparisons with LFB on the datasets in the original paper,  and  (ii) application to more standard datasets.  Further, the "long-term" memory design is a misleading name, as the length of the context is limited (unlike in LSTMs where at least theoretically, the sequence length is not limited) -- I would strongly urge the authors to change the name of the proposed approach.
> >
> > Given the otherwise strong results on the datasets considered in this paper, I am inclined to maintain my original rating.

---

> > > ### Author Response · Authors · 2021-08-31
> > > **Re: Final Review**
> > >
> > > Considering the limited time of rebuttal and the changes required for applying our proposed model to other video understanding tasks, such as video classification, we plan to explore this direction as our future work. We sincerely thank you for your suggestions making the paper better.

---

### Official Review · Reviewer_Zx4R · 2021-07-22

**Rating:** 8
**Confidence:** 5

**Summary:**

This paper proposed a transformer-based architecture for online action detection. The architecture contains a long-term memory that stores information from the past 8 minutes and a short-term memory that processes the recent 8 seconds. Some techniques are used to reduce the complexity of transformer operations. The proposed method gets good results on THUMOS’14, TVSeries and HACS datasets.

**Ethical Concerns:**

There is no major ethical concern.

**Limitations And Societal Impact:**

Generally, the paper is good. Two limitations (suggestions) below:

* In the online inference setting (section 3.6), the current visual feature predicts actions conditioning on both the short and long memory. But in section 3.5 (Training LSTR), it is said a directional attention mask is used such that each element in the short-term memory (current -- 8s ago) can have an action prediction. In the training case, the earlier visual features are having less short-term memory, e.g. the earliest feature at 8s ago is not utilizing any short-term memory at all - which is different with inference. I wonder if this directional attention-based training setting is the most efficient.

* The purpose of the first transformer decoder unit is to apply a dimension reduction for the long-term memory. Have authors tried other simpler alternative designs, like a projection layer?

**Main Review:**

Using transformers to model long-short term temporal information is a novel architecture design, and I think the attention mechanism is suitable for retrieving information from long-short term memories. The proposed memory compression method for complexity reduction is a useful tool for long-term video modelling.

Overall, the experiments are well-conducted to show the effect of design choices. The analysis of runtime in the supplementary material is valuable for future research to follow.

The submission is clearly written. As a suggestion, Fig 1 can be made more concise by avoiding repeating the video stream.

The proposed LSTR model achieves state-of-the-art results on online action detection tasks on three common datasets, showing the effectiveness of the transformers and the long-short term design.


**Time Spent Reviewing:**

5

---

> ### Author Response · Authors · 2021-08-10
> **Response to Reviewer Zx4R**
>
> Thank you for your review. We appreciate that you find our architecture design novel, and our paper clearly written. We address your concerns below.
>
> **Q1. “As a suggestion, Fig 1 can be made more concise ... the video stream.”**
>
> **Answer:** Thanks for this suggestion. We will further polish Fig 1 to be more concise in the revision.
>
> **Q2. “In the online inference setting (section 3.6) ... directional attention mask ... the most efficient.”**
>
> **Answer:** Thanks for the question. We incorporate the directional attention mask for the following reasons. (1) During training, randomly adjusting the short-term memory length for each frame can serve as data augmentation, so that LSTR does not overfit to the pre-defined short-term memory length. Using the directional attention mask is a way to efficiently and effectively implement this idea. For each training epoch, we randomly choose the start location of each video, and each frame has a random chance to access different short-term memory lengths. In practice, we found this data augmentation significantly facilities the training of LSTR --- the model experienced severe overfitting problems when training with a fixed short-term length. (2) The directional attention mask also ensures that there is no violation of the online setting (i.e., prediction of the present cannot see the future) in the short-term memory. With this design, we can efficiently provide supervision on every frame in the short-term memory (please refer to L179-L183). During inference, we run LSTR frame-by-frame along the streaming video, and only take the outputs from the last (i.e., the current) time step for classification, thus every frame can utilize the entire short-term memory. We will clarify this in the revision.
>
> **Q3. “The purpose of the first transformer decoder unit ... like a projection layer?”**
>
> **Answer:** As suggested, we test to replace the first transformer decoder unit with a projection layer, while keeping other model designs the same. The projection layer mainly consists of a learnable matrix of size $n_0 \times m_L$, and aims to compress the long-term memory along the temporal dimension. Thus, the long-term memory of size $m_L \times C$ can be compressed into size of $n_0 \times C$ by matrix multiplication. We also apply MLP layers after the projection layer to ensure that its number of model parameters is comparable to LSTR. We evaluate this design on THUMOS with $m_S$ as 8s and $m_L$ from 8s to 512s. Results are shown in the following table. We observe that modeling long-term memory with this simple projection layer slightly outperforms the design without using long-term memory (65.7% mAP). However, using LSTR’s attention-based compression stably outperforms this baseline and the performance gap increases when $m_L$ is getting larger.
>
> | | 8s | 16s | 32s | 64s | 128s | 256s | 512s |
> | :---- | :----: | :----: | :----: | :----: | :----: | :----: | :----: |
> | Using Projection Layer | 66.2 | 67.1 | 67.4 | 67.7 | 67.5 | 67.2 | 66.9 |
> | LSTR | 66.6 | 67.8 | 68.2 | 68.8 | 69.2 | 69.4 | 69.5 |

---

> > ### Comment · Reviewer_Zx4R · 2021-08-22
> > **Second-round comments**
> >
> > I read all the reviews and the authors' responses. The authors have clearly addressed my questions in Q2 & Q3 above. For Q3 they provide another experiment to verify their design choice. I will suggest an acceptance for this paper and keep my previous rating.

---

### Decision · Program_Chairs · 2021-09-27

**Decision:**

Accept (Spotlight)

**Comment:**

The LSTR model is original and well explained and explored in the paper. It was well received by all reviewers, and the rebuttal has addressed most of the concerns raised.

One outstanding issue is that the model is not applied to several of the current datasets and tasks of interest (e.g. AVA, EPIC-Kitchens) and this will somewhat limit interest in it. The authors are encouraged to address this in the final version of the paper.